# EPISODIC CURIOSITY THROUGH REACHABILITY

**Nikolay Savinov**[*1]    **Anton Raichuk**[*1]    **Raphaël Marinier**[*1]    **Damien Vincent**[*1]
**Marc Pollefeys**[3]    **Timothy Lillicrap**[2]    **Sylvain Gelly**[1]

[1]Google Brain,  [2]DeepMind,  [3]ETH Zürich

## ABSTRACT

Rewards are sparse in the real world and most of today's reinforcement learning algorithms struggle with such sparsity. One solution to this problem is to allow the agent to create rewards for itself — thus making rewards dense and more suitable for learning. In particular, inspired by curious behaviour in animals, observing something novel could be rewarded with a bonus. Such bonus is summed up with the real task reward — making it possible for RL algorithms to learn from the combined reward. We propose a new curiosity method which uses episodic memory to form the novelty bonus. To determine the bonus, the current observation is compared with the observations in memory. Crucially, the comparison is done based on how many environment steps it takes to reach the current observation from those in memory — which incorporates rich information about environment dynamics. This allows us to overcome the known "couch-potato" issues of prior work — when the agent finds a way to instantly gratify itself by exploiting actions which lead to hardly predictable consequences. We test our approach in visually rich 3D environments in *VizDoom*, *DMLab* and *MuJoCo*. In navigational tasks from *VizDoom* and *DMLab*, our agent outperforms the state-of-the-art curiosity method ICM. In *MuJoCo*, an ant equipped with our curiosity module learns locomotion out of the first-person-view curiosity only. The code is available at https://github.com/google-research/episodic-curiosity.

## 1 INTRODUCTION

Many real-world tasks have sparse rewards. For example, animals searching for food may need to go many miles without any reward from the environment. Standard reinforcement learning algorithms struggle with such tasks because of reliance on simple action entropy maximization as a source of exploration behaviour.

Multiple approaches were proposed to achieve better explorative policies. One way is to give a reward bonus which facilitates exploration by rewarding novel observations. The reward bonus is summed up with the original task reward and optimized by standard RL algorithms. Such an approach is motivated by neuroscience studies of animals: an animal has an ability to reward itself for something novel – the mechanism biologically built into its dopamine release system. How exactly this bonus is formed remains an open question.

Many modern curiosity formulations aim at maximizing "surprise" — inability to predict the future. This approach makes perfect sense but, in fact, is far from perfect. To show why, let us consider a thought experiment. Imagine an agent is put into a 3D maze. There is a precious goal somewhere in the maze which would give a large reward. Now, the agent is also given a remote control to a TV and can switch the channels. Every switch shows a random image (say, from a fixed set of images). The curiosity formulations which optimize surprise would rejoice because the result of the channel switching action is unpredictable. The agent would be drawn to the TV instead of looking for a goal in the environment (this was indeed observed in (Burda et al., 2018a)). So, should we call the channel switching behaviour curious? Maybe, but it is unproductive for the original sparse-reward goal-reaching task. What would be a definition of curiosity which does not suffer from such "couch-potato" behaviour?

We propose a new curiosity definition based on the following intuition. If the agent knew the observation after changing a TV channel is only one step away from the observation before doing that — it probably would not be so interesting to change the channel in the first place (too easy). This

---

[*]Shared first authorship.

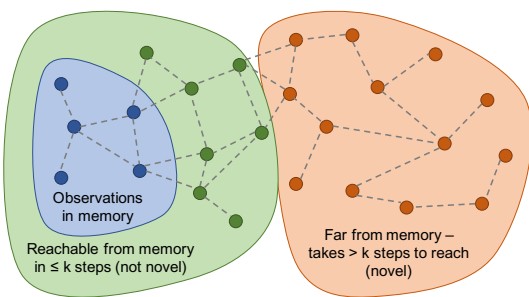

Figure 1: We define novelty through reach-ability. The nodes in the graph are observa-tions, the edges — possible transitions. The blue nodes are already in memory, the green nodes are reachable from the memory within $k = 2$ steps (not novel), the orange nodes are further away — take more than $k$ steps to reach (novel). In practice, the full possible transition graph is not available, so we train a neural network approximator to predict if the distance in steps between observations is larger or smaller than $k$.

intuition can be formalized as giving a reward only for those observations which take some effort to reach (outside the already explored part of the environment). The effort is measured in the number of environment steps. To estimate it we train a neural network approximator: given two observations, it would predict how many steps separate them. The concept of novelty via reachability is illustrated in Figure 1. To make the description above practically implementable, there is still one piece miss-ing though. For determining the novelty of the current observation, we need to keep track of what was already explored in the environment. A natural candidate for that purpose would be episodic memory: it stores instances of the past which makes it easy to apply the reachability approximator on pairs of current and past observations.

Our method works as follows. The agent starts with an empty memory at the beginning of the episode and at every step compares the current observation with the observations in memory to determine novelty. If the current observation is indeed novel — takes more steps to reach from observations in memory than a threshold — the agent rewards itself with a bonus and adds the current observation to the episodic memory. The process continues until the end of the episode, when the memory is wiped clean.

We benchmark our method on a range of tasks from visually rich 3D environments *VizDoom*, *DM-Lab* and *MuJoCo*. We conduct the comparison with other methods — including the state-of-the-art curiosity method ICM (Pathak et al., 2017) — under the same budget of environment interactions. First, we use the *VizDoom* environments from prior work to establish that our re-implementation of the ICM baseline is correct — and also demonstrate at least 2 times faster convergence of our method with respect to the baseline. Second, in the randomized procedurally generated environments from *DMLab* our method turns out to be more robust to spurious behaviours than the method ICM: while the baseline learns a persistent firing behaviour in navigational tasks (thus creating interesting pic-tures for itself), our method learns a reasonable explorative behaviour. In terms of quantitative evaluation, our method reaches the goal at least 2 times more often in the procedurally generated test levels in *DMLab* with a very sparse reward. Third, when comparing the behaviour of the agent in the complete absence of rewards, our method covers at least 4 times more area (measured in discrete $(x, y)$ coordinate cells) than the baseline ICM. Fourth, we demonstrate that our curiosity bonus does not significantly deteriorate performance of the plain PPO algorithm (Schulman et al., 2017) in two tasks with dense reward in *DMLab*. Finally, we demonstrate that an ant in a *MuJoCo* environment can learn locomotion purely from our curiosity reward computed based on the first-person view.

## 2 EPISODIC CURIOSITY

We consider an agent which interacts with an environment. The interactions happen at discrete time steps over the episodes of limited duration $T$. At each time step $t$, the environment provides the agent with an observation $\mathbf{o}_t$ from the observational space $\mathcal{O}$ (we consider images), samples an action $a_t$ from a set of actions $\mathcal{A}$ using a probabilistic policy $\pi(\mathbf{o}_t)$ and receives a scalar reward $r_t \in \mathbb{R}$ together with the new observation $\mathbf{o}_{t+1}$ and an end-of-episode indicator. The goal of the agent is to optimize the expectation of the discounted sum of rewards during the episode $S = \sum_t \gamma^t r_t$.

In this work we primarily focus on the tasks where rewards $r_t$ are sparse — that is, zero for most of the time steps $t$. Under such conditions commonly used RL algorithms (e.g., PPO Schulman et al. (2017)) do not work well. We further introduce an episodic curiosity (EC) module which alleviates this problem. The purpose of this module is to produce a reward bonus $b_t$ which is further summed

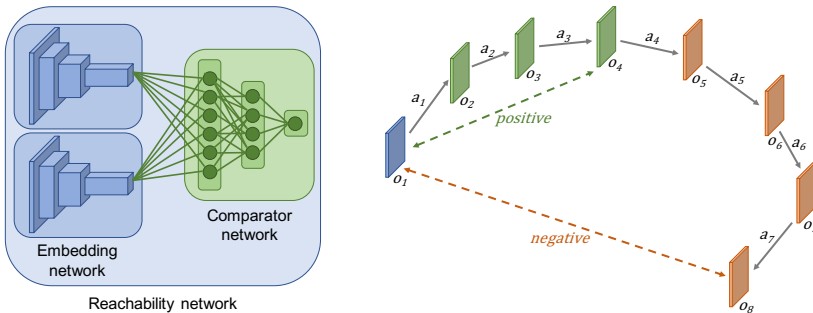

Figure 2: Left: siamese architecture of reachability (R) network. Right: R-network is trained based on a sequence of observations that the agent encounters while acting. The temporally close (within threshold) pairs of observations are positive examples, while temporally far ones — negatives.

up with the task reward $r_t$ to give an augmented reward $\widehat{r}_t = r_t + b_t$. The augmented reward has a nice property from the RL point of view — it is a dense reward. Learning with such reward is faster, more stable and often leads to better final performance in terms of the cumulative task reward $S$.

In the following section we describe the key components of our episodic curiosity module.

## 2.1 EPISODIC CURIOSITY MODULE

The episodic curiosity (EC) module takes the current observation $\mathbf{o}$ as input and produces a reward bonus $b$. The module consists of both parametric and non-parametric components. There are two parametric components: an embedding network $E : \mathcal{O} \to \mathbb{R}^n$ and a comparator network $C : \mathbb{R}^n \times \mathbb{R}^n \to [0, 1]$. Those parametric components are trained together to predict reachability as parts of the reachability network — shown in Figure 2. There are also two non-parametric components: an episodic memory buffer $\mathbf{M}$ and a reward bonus estimation function $B$. The high-level overview of the system is shown in Figure 3. Next, we give a detailed explanation of all the components.

**Embedding and comparator networks.** Both networks are designed to function jointly for estimating within-$k$-step-reachability of one observation $\mathbf{o}_i$ from another observation $\mathbf{o}_j$ as parts of a reachability network $R(\mathbf{o}_i, \mathbf{o}_j) = C(E(\mathbf{o}_i), E(\mathbf{o}_j))$. This is a siamese architecture similar to (Zagoruyko & Komodakis, 2015). The architecture is shown in Figure 2. R-network is a classifier trained with a logistic regression loss: it predicts values close to $0$ if probability of two observations being reachable from one another within $k$ steps is low, and values close to $1$ when this probability is high. Inside the episodic curiosity the two networks are used separately to save up computation and memory.

**Episodic memory.** The episodic memory buffer $\mathbf{M}$ stores embeddings of past observations from the current episode, computed with the embedding network $E$. The memory buffer has a limited capacity $K$ to avoid memory and performance issues. At every step, the embedding of the current observation might be added to the memory. What to do when the capacity is exceeded? One solution we found working well in practice is to substitute a random element in memory with the current element. This way there are still more fresh elements in memory than older ones, but the older elements are not totally neglected.

**Reward bonus estimation module.** The purpose of this module is to check for reachable observations in memory and if none is found — assign larger reward bonus to the current time step. The check is done by comparing embeddings in memory to the current embedding via comparator network. Essentially, this check insures that no observation in memory can be reached by taking only a few actions from the current state — our characterization of novelty.

## 2.2 BONUS COMPUTATION ALGORITHM.

At every time step, the current observation $\mathbf{o}$ goes through the embedding network producing the embedding vector $\mathbf{e} = E(\mathbf{o})$. This embedding vector is compared with the stored embeddings in the memory buffer $\mathbf{M} = \langle \mathbf{e}_1, \ldots, \mathbf{e}_{|\mathbf{M}|} \rangle$ via the comparator network $C$ where $|\mathbf{M}|$ is the current number of elements in memory. This comparator network fills the reachability buffer with values

$$c_i = C(\mathbf{e}_i, \mathbf{e}), \quad i = 1, |\mathbf{M}|. \tag{1}$$

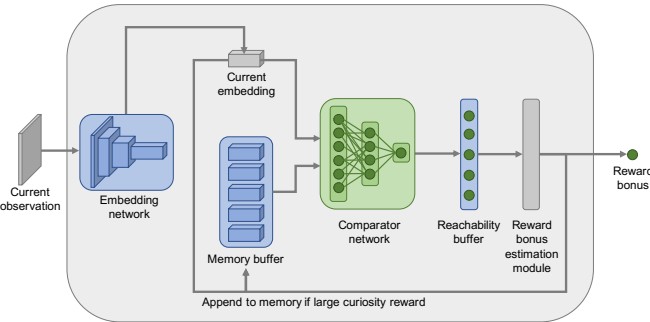

Figure 3: The use of episodic curiosity (EC) module for reward bonus computation. The module take a current observation as input and computes a reward bonus which is higher for novel observations. This bonus is later summed up with the task reward and used for training an RL agent.

Then the similarity score between the memory buffer and the current embedding is computed from the reachability buffer as (with a slight abuse of notation)

$$C(\mathbf{M}, \mathbf{e}) = F\left(c_1, \ldots, c_{|\mathbf{M}|}\right) \in [0, 1]. \tag{2}$$

where the aggregation function $F$ is a hyperparameter of our method. Theoretically, $F = \max$ would be a good choice, however, in practice it is prone to outliers coming from the parametric embedding and comparator networks. Empirically, we found that 90-th percentile works well as a robust substitute to maximum.

As a curiosity bonus, we take

$$b = B(\mathbf{M}, \mathbf{e}) = \alpha(\beta - C(\mathbf{M}, \mathbf{e})), \tag{3}$$

where $\alpha \in \mathbb{R}^+$ and $\beta \in \mathbb{R}$ are hyperparameters of our method. The value of $\alpha$ depends on the scale of task rewards — we will discuss how to select it in the experimental section. The value of $\beta$ determines the sign of the reward — and thus could bias the episodes to be shorter or longer. Empirically, $\beta = 0.5$ works well for fixed-duration episodes, and $\beta = 1$ is preferred if an episode could have variable length.

After the bonus computation, the observation embedding is added to memory if the bonus $b$ is larger than a novelty threshold $b_{novelty}$. This check is necessary for the following reason. If every observation embedding is added to the memory buffer, the observation from the current step will always be reachable from the previous step. Thus, the reward would never be granted. The threshold $b_{novelty}$ induces a discretization in the embedding space. Intuitively, this makes sense: only "distinct enough" memories are stored. As a side benefit, the memory buffer stores information with much less redundancy. We refer the reader to the video[1] which visualizes the curiosity reward bonus and the memory state during the operation of the algorithm.

### 2.3 REACHABILITY NETWORK TRAINING

If the full transition graph in Figure 1 was available, there would be no need of a reachability network and the novelty could be computed analytically through the shortest-path algorithm. However, normally we have access only to the sequence of observations which the agent receives while acting. Fortunately, as suggested by (Savinov et al., 2018), even a simple observation sequence graph could still be used for training a reasonable approximator to the real step-distance. This procedure is illustrated in Figure 2. This procedure takes as input a sequence of observations $\mathbf{o}_1, \ldots, \mathbf{o}_N$ and forms pairs from those observations. The pairs $(\mathbf{o}_i, \mathbf{o}_j)$ where $|i - j| \le k$ are taken as positive (reachable) examples while the pairs with $|i - j| > \gamma k$ become negative examples. The hyperparameter $\gamma$ is necessary to create a gap between positive and negative examples. In the end, the network is trained with logistic regression loss to output the probability of the positive (reachable) class.

In our work, we have explored two settings for training a reachability network: using a random policy and together with the task-solving policy (online training). The first version generally follows the training protocol proposed by (Savinov et al., 2018). We put the agent into exactly the same

---

[1]https://youtu.be/mphIRR6VsbM

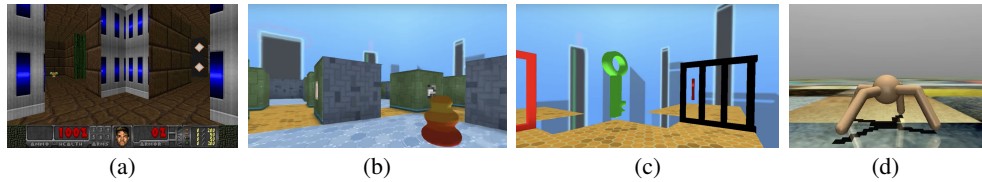

|  |  |  |  |
|:-:|:-:|:-:|:-:|
| (a) | (b) | (c) | (d) |

Figure 4: Examples of tasks considered in our experiments: (a) *VizDoom* static maze goal reaching, (b) *DMLab* randomized maze goal reaching, (c) *DMLab* key-door puzzle, (d) *MuJoCo* ant locomotion out of first-person-view curiosity.

conditions where it will be eventually tested: same episode duration and same action set. The agent takes random actions from the action set. Given the environment interaction budget (2.5M 4-repeated steps in *DMLab*, 300K 4-repeated steps in *VizDoom*), the agent fills in the replay buffer with observations coming from its interactions with the environment, and forms training pairs by sampling from this replay buffer randomly. The second version collects the data on-policy, and re-trains the reachability network every time after a fixed number of environment interactions is performed. We provide the details of R-network training in the supplementary material.

## 3 EXPERIMENTAL SETUP

We test our method in multiple environments from *VizDoom* (Kempka et al., 2016), *DMLab* (Beattie et al., 2016) and *MuJoCo* (Todorov et al., 2012; Schulman et al., 2015). The experiments in *Viz-Doom* allow us to verify that our re-implementation of the previous state-of-the-art curiosity method ICM (Pathak et al., 2017) is correct. The experiments in *DMLab* allow us to extensively test the generalization of our method as well as baselines — *DMLab* provides convenient procedural level generation capabilities which allows us to train and test RL methods on hundreds of levels. The experiments in *MuJoCo* allow us to show the generality of our method. Due to space limits, the *MuJoCo* experiments are described in the supplementary material. The examples of tasks are shown in Figure 4.

**Environments.** Both *VizDoom* and *DMLab* environments provide rich maze-like 3D environments. The observations are given to the agent in the form of images. For *VizDoom*, we use $84 \times 84$ grayscale images as input. For *DMLab*, we use $84 \times 84$ RGB images as input. The agent operates with a discrete action set which comprises different navigational actions. For *VizDoom*, the standard action set consists of 3 actions: move forward, turn left/right. For *DMLab*, it consists of 9 actions: move forward/backward, turn left/right, strafe left/right, turn left/right+move forward, fire. For both *VizDoom* and *DMLab* we use all actions with a repeat of 4, as typical in the prior work. We only use RGB input of the provided RGBD observations and remove all head-on display information from the screen, leaving only the plain first-person view images of the maze. The rewards and episode durations differ between particular environments and will be further specified in the corresponding experimental sections.

**Basic RL algorithm.** We choose the commonly used PPO algorithm from the open-source implementation[2] as our basic RL algorithm. The policy and value functions are represented as CNNs to reduce number of hyperparameters — LSTMs are harder to tune and such tuning is orthogonal to the contribution of the paper. We apply PPO to the sum of the task reward and the bonus reward coming from specific curiosity algorithms. The hyperparameters of the PPO algorithm are given in the supplementary material. We use only two sets of hyperparameters: one for all *VizDoom* environments and the other one for all *DMLab* environments.

**Baseline methods.** The simplest baseline for our approach is just the basic RL algorithm applied to the task reward. As suggested by the prior work and our experiments, this is a relatively weak baseline in the tasks where reward is sparse.

As the second baseline, we take the state-of-the-art curiosity method ICM (Pathak et al., 2017). As follows from the results in (Pathak et al., 2017; Fu et al., 2017), ICM is superior to methods VIME (Houthooft et al., 2016), #Exploration (Tang et al., 2017) and $EX^2$ (Fu et al., 2017) on the curiosity tasks in visually rich 3D environments.

---

[2] https://github.com/openai/baselines

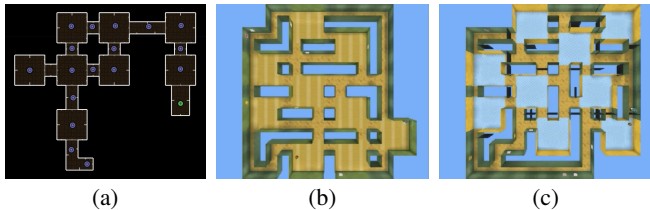

|  (a)  |  (b)  |  (c)  |

Figure 5: Examples of maze types used in our experiments: (a) *VizDoom* static maze goal reaching, (b) *DMLab* randomized maze goal reaching, (c) *DMLab* randomized maze goal reaching with doors.

Finally, as a sanity check, we introduce a novel baseline method which we call Grid Oracle. Since we can access current $(x, y)$ coordinates of the agent in all environments, we are able to directly discretize the world in 2D cells and reward the agent for visiting as many cells as possible during the episode (the reward bonus is proportional to the number of cells visited). At the end of the episode, cell visit counts are zeroed. The reader should keep in mind that this baseline uses privileged information not available to other methods (including our own method EC). While this privileged information is not guaranteed to lead to success in any particular RL task, we do observe this baseline to perform strongly in many tasks, especially in complicated *DMLab* environments. The Grid Oracle baseline has two hyperparameters: the weight for combining Grid Oracle reward with the task reward and the cell size.

**Hyperparameter tuning.** As *DMLab* environments are procedurally generated, we perform tuning on the validation set, disjoint with the training and test sets. The tuning is done on one of the environments and then the same hyperparameters are re-used for all other environments. *VizDoom* environments are not procedurally generated, so there is no trivial way to have proper training/validation/test splits — so we tune on the same environment (as typical in the prior RL work for the environments without splits). When tuning, we consider the mean final reward of 10 training runs with the same set of hyperparameters as the objective — thus we do not perform any seed tuning. All hyperparameter values are listed in the supplementary material. Note that although bonus scalar $\alpha$ depends on the range of task rewards, the environments in *VizDoom* and *DMLab* have similar ranges within each platform — so our approach with re-using $\alpha$ for multiple environments works.

## 4 EXPERIMENTS

In this section, we describe the specific tasks we are solving and experimental results for all considered methods on those tasks. There are 4 methods to report: PPO, PPO + ICM, PPO + Grid Oracle and PPO + EC (our method). First, we test static-maze goal reaching in *VizDoom* environments from prior work to verify that our baseline re-implementation is correct. Second, we test the goal-reaching behaviour in procedurally generated mazes in *DMLab*. Third, we train no-reward (pure curiosity) maze exploration on the levels from *DMLab* and report Grid Oracle reward as an approximate measure of the maze coverage. Finally, we demonstrate that our curiosity bonus does not significantly deteriorate performance in two dense reward tasks in *DMLab*. All the experiments were conducted under the same environment interaction budget for all methods (R-network pre-training is included in this budget). The videos of all trained agents in all environments are available online[3].

For additional experiments we refer the reader to the supplementary material: there we show that R-network can successfully generalize between environments, demonstrate stability of our method to hyperparameters and present an ablation study.

### 4.1 STATIC MAZE GOAL REACHING.

The goal of this experiment is to verify our re-implementation of the baseline method is correct. We use the MyWayHome task from *VizDoom*. The agent has to reach the goal in a static 3D maze in the time limit of $525$ 4-repeated steps (equivalent to 1 minute). It only gets a reward of $+1$ when it reaches the goal (episode ends at that moment), the rest of the time the reward is zero.

The task has three sub-tasks (following the setup in (Pathak et al., 2017)): "Dense", "Sparse" and "Very Sparse". The layout of the maze is demonstrated in Figure 5(c). The goal is always at the same room but the starting points are different in those sub-tasks. For the "Dense" subtask, the

---
[3]https://sites.google.com/view/episodic-curiosity

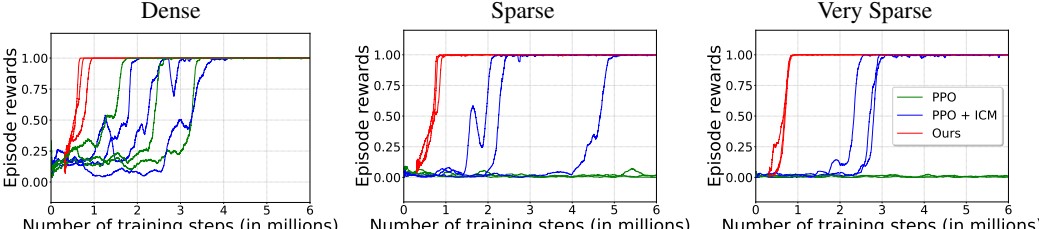

Figure 6: Task reward as a function of training step for *VizDoom* tasks. Higher is better. We use the offline version of our algorithm and shift the curves for our method by the number of environment steps used to train R-network — so the comparison is fair. We run every method with a repeat of 3 (same as in prior work (Pathak et al., 2017)) and show all runs. No seed tuning is performed.

agent starts in one of the random locations in the maze, some of which are close to the goal. In this sub-task, the reward is relatively dense (hence the name): the agent is likely to bump into the goal by a short random walk. Thus, this is an easy task even for standard RL methods. The other two sub-tasks are harder: the agent starts in a medium-distant room from the goal ("Sparse") or in a very distant room ("Very Sparse"). Those tasks are hard for standard RL algorithms because the probability of bumping into a rewarding state by a random walk is very low.

The training curves are shown in Figure 6. By analysing them, we draw a few conclusions. First, our re-implementation of the ICM baseline is correct and the results are in line with those published in (Pathak et al., 2017). Second, our method works on-par with the ICM baseline in terms of final performance, quickly reaching 100% success rate in all three sub-tasks. Finally, in terms of convergence speed, our algorithm is significantly faster than the state-of-the-art method ICM — our method reaches 100% success rate at least 2 times faster. Note that to make the comparison of the training speed fair, we shift our training curves by the environment interaction budget used for training R-network.

### 4.2 PROCEDURALLY GENERATED RANDOM MAZE GOAL REACHING.

In this experiment we aim to evaluate maze goal reaching task generalization on a large scale. We train on hundreds of levels and then test also on hundreds of hold-out levels. We use "Explore Goal Locations Large" (we will denote it "Sparse") and "Explore Obstructed Goals Large" (we will denote it "Sparse + Doors") levels in the *DMLab* simulator. In those levels, the agent starts in a random location in a randomly generated maze (both layout and textures are randomized at the beginning of the episode). Within the time limit of 1800 4-repeated steps (equivalent to 2 minutes), the agent has to reach the goal as many times as possible. Every time it reaches a goal, it is respawned into another random location in the maze and has to go to the goal again. Every time the goal is reached, the agent gets a reward +10, the rest of the time the reward is zero. The second level is a variation of the first one with doors which make the paths in the maze longer. The layouts of the levels are demonstrated in Figure 5(b,c).

We found out that the standard task "Sparse" is actually relatively easy even for the plain PPO algorithm. The reason is that the agent starting point and the goal are sampled on the map independently of each other — and sometimes both happen to be in the same room which simplifies the task. To test the limits of the algorithms, we create a gap between the starting point and the goal which eliminates same-room initialization. We report the results for both the original task "Sparse" and its harder version "Very Sparse". Thus, there are overall three tasks considered in this section: "Sparse", "Very Sparse" and "Sparse + Doors".

The results demonstrate that our method can reasonably adapt to ever-changing layouts and textures — see Table 1 and training curves in Figure 7. We outperform the baseline method ICM in all three environments using the same environment interaction budget of 20M 4-repeated steps. The environment "Sparse" is relatively easy and all methods work reasonably. In the "Very Sparse" and "Sparse + Doors" settings our advantage with respect to PPO and ICM is more clear. On those levels, the visual inspection of the ICM learnt behaviour reveals an important property of this method: it is confused by the firing action and learns to entertain itself by firing until it runs out of ammunition. A similar finding was reported in a concurrent work (Burda et al., 2018a): the agent was given an action which switched the content on a TV screen in a maze, along with the movement actions. Instead of moving, the agent learns to switch channels forever. While one might intuitively

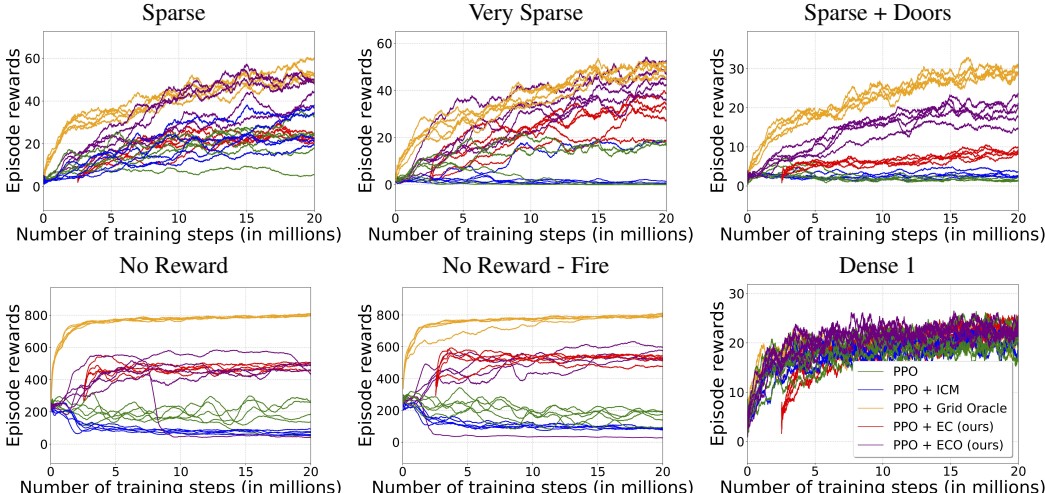

Figure 7: Reward as a function of training step for *DMLab* tasks. Higher is better. "ECO" stands for the online version of our method, which trains R-network and the policy at the same time. We run every method 30 times and show 5 randomly selected runs. No seed tuning is performed.

accept such "couch-potato" behaviour in intelligent creatures, it does not need to be a consequence of curious behaviour. In particular, we are not observing such dramatic firing behaviour for our curiosity formulation: according to Figure 1, an observation after firing is still one step away from the one before firing, so it is not novel (note that firing still could happen in practice because of the entropy term in PPO). Thus, our formulation turns out to be more robust than ICM's prediction error in this scenario. Note that we do not specifically look for an action set which breaks the baseline — just use the standard one for *DMLab*, in line with the prior work (e.g., (Espeholt et al., 2018)).

The result of this experiment suggests to look more into how methods behave in extremely-sparse reward scenarios. The limiting case would be no reward at all — we consider it in the next section.

### 4.3 NO REWARD/AREA COVERAGE.

This experiment aims to quantitatively establish how good our method is in the scenario when no task reward is given. One might question why this scenario is interesting — however, before the task reward is found for the first time, the agent lives in the no-reward world. How it behaves in this case will also determine how likely it is to stumble into the task reward in the first place.

We use one of the *DMLab* levels — "Sparse" from the previous experiment. We modify the task to eliminate the reward and name the new task "No Reward". To quantify success in this task, we report the reward coming from Grid Oracle for all compared methods. This reward provides a discrete approximation to the area covered by the agent while exploring.

The training curves are shown in Figure 7 and the final test results in Table 1. The result of this experiment is that our method and Grid Oracle both work, while the ICM baseline is not working — and the qualitative difference in behaviour is bigger than in the previous experiments. As can be seen from the training curves, after a temporary increase, ICM quality actually decreases over time, rendering a sharp disagreement between the prediction-error-based bonus and the area coverage metric. By looking at the video[3], we observe that the firing behaviour of ICM becomes even more prominent, while our method still shows reasonable exploration.

Finally, we try to find out if the ICM baseline behaviour above is due to the firing action only. Could it learn exploration of randomized mazes if the Fire action is excluded from the action set? For that purpose, we create a new version of the task — we call it "No Reward - Fire". This task demonstrates qualitatively similar results to the one with the full action set — see Table 1. By looking at the videos[3], we hypothesise that the agent can most significantly change its current view when it is close to the wall — thus increasing one-step prediction error — so it tends to get stuck near "interesting" diverse textures on the walls.

The results suggest that in an environment completely without reward, the ICM method will exhaust its curiosity very quickly — passing through a sharp peak and then degrading into undesired be-

Table 1: Reward in *DMLab* tasks (mean ± std) for all compared methods. Higher is better. "ECO" stands for the online version of our method, which trains R-network and the policy at the same time. We report Grid Oracle reward in tasks with no reward. The Grid Oracle method is given for reference — it uses privileged information unavailable to other methods. Results are averaged over 30 random seeds. No seed tuning is performed.

| Method | Sparse | Very Sparse | Sparse+Doors | No Reward | No Reward - Fire | Dense 1 | Dense 2 |
|---|---|---|---|---|---|---|---|
| PPO | 27.0 ± 5.1 | 8.6 ± 4.3 | 1.5 ± 0.1 | 191 ± 12 | 217 ± 19 | 22.8 ± 0.5 | 9.41 ± 0.02 |
| PPO + ICM | 23.8 ± 2.8 | 11.2 ± 3.9 | 2.7 ± 0.2 | 72 ± 2 | 87 ± 3 | 20.9 ± 0.6 | 9.39 ± 0.02 |
| PPO + EC (ours) | 26.2 ± 1.9 | 24.7 ± 2.2 | 8.5 ± 0.6 | **475 ± 8** | **492 ± 10** | 19.9 ± 0.7 | 9.53 ± 0.03 |
| PPO + ECO (ours) | **41.6 ± 1.7** | **40.5 ± 1.1** | **19.8 ± 0.5** | 472 ± 18 | 457 ± 32 | **22.9 ± 0.4** | **9.60 ± 0.02** |
| PPO + Grid Oracle | 56.7 ± 1.3 | 54.3 ± 1.2 | 29.4 ± 0.5 | 796 ± 2 | 795 ± 3 | 20.9 ± 0.6 | 8.97 ± 0.04 |

haviour. This observation raises concerns: what if ICM passes the peak before it reaches the first task reward in the cases of real tasks? Supposedly, it would require careful tuning per-game. Furthermore, in some cases, it would take a lot of time with a good exploration behaviour to reach the first reward, which would require to stay at the top performance for longer — which is problematic for the ICM method but still possible for our method.

### 4.4 DENSE REWARD TASKS.

A desirable property of a good curiosity bonus is to avoid hurting performance in dense-reward tasks (in addition to improving performance for sparse-reward tasks). We test this scenario in two levels in the *DMLab* simulator: "Rooms Keys Doors Puzzle" (which we denote "Dense 1") and "Rooms Collect Good Objects Train" (which we denote "Dense 2"). In the first task, the agent has to collect keys and reach the goal object behind a few doors openable by those keys. The rewards in this task are rather dense (key collection/door opening is rewarded). In the second task the agent has to collect good objects (give positive reward) and avoid bad objects (give negative reward). The episode lasts for 900 4-repeated steps (equivalent to 1 minute) in both tasks.

The results show that our method indeed does not significantly deteriorate performance of plain PPO in those dense-reward tasks — see Table 1. The training curves for "Dense 1" are shown in Figure 7 and for "Dense 2" — in the supplementary material. Note that we use the same bonus weight in this task as in other *DMLab* tasks before. All methods work similarly besides the Grid Oracle in the "Dense 2" task — which performs slightly worse. Video inspection[3] reveals that Grid Oracle — the only method which has ground-truth knowledge about area it covers during training — sometimes runs around excessively and occasionally fails to collect all good objects.

## 5 DISCUSSION

Our method is at the intersection of multiple topics: curiosity, episodic memory and temporal distance prediction. In the following, we discuss the relation to the prior work on those topics.

**Curiosity in visually rich 3D environments.** Recently, a few works demonstrated the possibility to learn exploration behaviour in visually rich 3D environments like *DMLab* (Beattie et al., 2016) and *VizDoom* (Kempka et al., 2016). (Pathak et al., 2017) trains a predictor for the embedding of the next observation and if the reality is significantly different from the prediction — rewards the agent. In that work, the embedding is trained with the purpose to be a good embedding for predicting action taken between observations — unlike an earlier work (Stadie et al., 2015) which obtains an embedding from an autoencoder. It was later shown by (Burda et al., 2018a) that the perceptive prediction approach has a downside — the agent could become a "couch-potato" if given an action to switch TV channels. This observation is confirmed in our experiments by observing a persistent firing behaviour of the ICM baseline in the navigational tasks with very sparse or no reward. By contrast, our method does not show this behaviour. Another work (Fu et al., 2017) trains a temporal distance predictor and then uses this predictor to establish novelty: if the observation is easy to classify versus previous observations, it is novel. This method does not use episodic memory, however, and the predictor is used in way which is different from our work.

**General curiosity.** Curiosity-based exploration for RL has been extensively studied in the literature. For an overview, we refer the reader to the works (Oudeyer & Kaplan, 2009; Oudeyer et al., 2007). The most common practical approaches could be divided into three branches: prediction-

error-based, count-based and goal-generation-based. Since the prediction-based approaches were discussed before, in the following we focus on the latter two branches.

The count-based approach suggests to keep visit counts for observations and concentrate on visiting states which has been rarely visited before — which bears distant similarity to how we use episodic memory. This idea is natural for discrete observation spaces and has solid theoretical foundations. Its extension to continuous observation spaces is non-trivial, however. The notable step in this direction was taken by works (Bellemare et al., 2016; Ostrovski et al., 2017) which introduce a trained observation density model which is later converted to a function behaving similarly to counts. The way conversion is done has some similarity to prediction-error-based approaches: it is the difference of the density in the example before and after training of this example which is converted to count. The experiments in the original works operate on Atari games (Bellemare et al., 2013) and were not benchmarked on visually rich 3D environments. Another approach (Tang et al., 2017) discretises the continuous observation space by hashing and then uses the count-based approach in this discretised space. This method is appealing in its simplicity, however, the experiments in (Pathak et al., 2017; Fu et al., 2017) show that it does not perform well in visually rich 3D environments. Another line of work, Novelty Search (Lehman & Stanley, 2011) and its recent follow-up (Conti et al., 2018), proposed maintaining an archive of behaviours and comparing current behaviour to those — however, the comparison is done by euclidean distance and behaviours are encoded using coordinates, while we learn the comparison function and only use pixels.

Finally, our concept of novelty through reachability is reminiscent of generating the goals which are reachable but not too easy — a well-studied topic in the prior work. The work (Held et al., 2017) uses a GAN to differentiate what is easy to reach from what is not and then generate goals at the boundary. Another work (Baranes & Oudeyer, 2013) defines new goals according to the expected progress the agent will make if it learns to solve the associated task. The recent work (Péré et al., 2018) learns an embedding for the goal space and then samples increasingly difficult goals from that space. In a spirit similar to those works, our method implicitly defines goals that are at least some fixed number of steps away by using the reachability network. However, our method is easier to implement than other goal-generation methods and quite general.

**Episodic memory.** Two recent works (Blundell et al., 2016; Pritzel et al., 2017) were inspired by the ideas of episodic memory in animals and proposed an approach to learn the functioning of episodic memory along with the task for which this memory is applied. Those works are more focused on repeating successful strategies than on exploring environments — and are not designed to work in the absence of task rewards.

**Temporal distance prediction.** The idea to predict the distance between video frames has been studied extensively. Usually this prediction is an auxiliary task for solving another problem. (Sermanet et al., 2017) trains an embedding such that closer in time frames are also closer in the embedding space. Multiple works (Fu et al., 2017; Savinov et al., 2018; Aytar et al., 2018) train a binary classifier for predicting if the distance in time between frames is within a certain threshold or not. While (Sermanet et al., 2017; Aytar et al., 2018) use only the embedding for their algorithms, (Fu et al., 2017; Savinov et al., 2018) also use the classifier trained together with the embedding. As mentioned earlier, (Fu et al., 2017) uses this classifier for density estimation instead of comparison to episodic memory. (Savinov et al., 2018) does compare to the episodic memory buffer but solves a different task — given an already provided exploration video, navigate to a goal — which is complementary to the task in our work.

## 6    CONCLUSION

In this work we propose a new model of curiosity based on episodic memory and the ideas of reachability. This allows us to overcome the known "couch-potato" issues of prior work and outperform the previous curiosity state-of-the-art method ICM in visually rich 3D environments from *VizDoom* and *DMLab*. Our method also allows a *MuJoCo* ant to learn locomotion purely out of first-person-view curiosity. In the future, we want to make policy aware of memory not only in terms of receiving reward, but also in terms of acting. Can we use memory content retrieved based on reachability to guide exploration behaviour in the test time? This could open opportunities to learn exploration in new tasks in a few-shot style — which is currently a big scientific challenge.

### ACKNOWLEDGMENTS

We would like to thank Olivier Pietquin, Alexey Dosovitskiy, Vladlen Koltun, Carlos Riquelme, Charles Blundell, Sergey Levine and Matthieu Geist for the valuable discussions about our work.

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

SUPPLEMENTARY MATERIAL

The supplementary material is organized as follows. First, we describe the *MuJoCo* locomotion experiments. Then we provide training details for R-network. After that, we list hyperparameter values and the details of hyperparameter search for all methods. Then we show experimental results which suggest that R-network can generalize between environments: we transfer one general R-network from all available *DMLab*30 levels to our tasks of interest and also transfer R-networks between single environments. After that, we present the results from a stability/ablation study which suggests our method is stable with respect to its most important hyperparameters and the components we used in the method are actually necessary for its performance (and measure their influence). Then we demonstrate the robustness of our method to the environments where every state has a stochastic next state. After that, we discuss computational considerations for our method. Finally, we provide the training curves for the "Dense 2" task in the main text.

## S1  *MuJoCo* ANT LOCOMOTION OUT OF FIRST-PERSON-VIEW CURIOSITY

Equipped with our curiosity module, a *MuJoCo* ant has learned[4] to move out of curiosity based on the first-person view[5].

First, let us describe the setup:

- Environment: the standard *MuJoCo* environment is a plane with a uniform or repetitive texture on it — nothing to be visually curious about. To fix that, we tiled the $400 \times 400$ floor into squares of size $4 \times 4$. Each tile is assigned a random texture from a set of 190 textures at the beginning of every episode. The ant is initialized at a random location in the $200 \times 200$ central square of the floor. The episode lasts for 1000 steps (no action repeat is used). If the $z$-coordinate of the center of mass of the ant is above 1.0 or below 0.2 — the episode ends prematurely (standard termination condition).

- Observation space: for computing the curiosity reward, we only use a first-person view camera mounted on the ant (that way we can use the same architecture of our curiosity module as in *VizDoom* and *DMLab*). For policy, we use the standard body features from Ant-v2 in gym-mujoco[6] (joint angles, velocities, etc.).

- Action space: standard continuous space from Ant-v2 in gym-mujoco.

- Basic RL solver: PPO (same as in the main text of the paper).

- Baselines: PPO on task reward, PPO on task reward plus constant reward 1 at every step as a trivial curiosity bonus (which we denote PPO+1, it optimizes for longer survival).

Second, we present quantitative results for the setting with no task reward after 10M training steps in Table S1 (the first row). Our method outperforms the baselines. As seen in the videos[7], PPO (random policy) dies quickly, PPO+1 survives for longer but does not move much and our method moves around the environment.

Additionally, we performed an experiment with an extremely sparse task reward — which we call "Escape Circle". The reward is given as follows: 0 reward inside the circle of radius 10, and starting from 10, we give a one-time reward of 1 every time an agent goes through a concentric circle of radius $10 + 0.5k$ (for integer $k \geq 0$). The results at 10M training steps are shown in Table S1 (the second row). Our method significantly outperforms the baselines (better than the best baseline by a factor of 10).

Finally, let us discuss the relation to some other works in the field of learning locomotion from intrinsic reward. The closest work in terms of task setup is the concurrent work (Burda et al., 2018a). The authors demonstrate slow motion[8] of the ant learned from pixel-based curiosity only.

---

[4]Behaviour learned by our method, third-person view: https://youtu.be/OYF9UcnEbQA

[5]Behaviour learned by our method, first-person view: https://youtu.be/klpDUdkv03k

[6]https://gym.openai.com/envs/Ant-v2/

[7]https://sites.google.com/view/episodic-curiosity

[8]https://youtu.be/l1FqtAHfJLI?t=90

Other works use state features (joint angles, velocities, etc.) for formulating intrinsic reward, not pixels — which is a different setup. One work in this direction is the concurrent work (Eysenbach et al., 2018) — which also contains a good overview of the literature on intrinsic reward from state features.

Table S1: Learning locomotion for *MuJoCo* Ant. For "No reward", the task reward is 0 (so plain PPO is a random policy), and Grid Oracle rewards are reported (with cell size 5). Results are averaged over 30 random seeds for "No reward" and over 10 random seeds for "Escape Circle". No seed tuning is performed.

| Task | PPO | PPO+1 | PPO + EC (ours) |
|------|-----|-------|-----------------|
| No Reward | $1.4 \pm 0.02$ | $1.7 \pm 0.06$ | $\mathbf{5.0 \pm 0.27}$ |
| Escape Circle | $0.59 \pm 0.54$ | $0.45 \pm 0.39$ | $\mathbf{6.53 \pm 3.57}$ |

## S2 REACHABILITY NETWORK TRAINING DETAILS

For training R-network, we use mini-batches of 64 observation pairs (matched within episodes). The training is run for 50K mini-batch iterations for *VizDoom* and 200K mini-batch iterations for *DMLab*. At the beginning of every pass through the buffer, we re-shuffle it. We use Adam optimizer with learning rate $10^{-4}$. The R-network uses a siamese architecture with two branches (see Figure 2 in the main text), each branch is Resnet-18 with 512 outputs, with a fully-connected network applied to the concatenated output of the branches. The fully-connected network has four hidden layers with 512 units, batch-normalization and ReLU is applied after each layer besides the last one, which is a softmax layer. Observations are RGB-images with resolution $160 \times 120$ pixels.

For online training of the R-network, we collect the experience and perform training every 720K 4-repeated environment steps. Every time the experience is collected, we make 10 epochs of training on this experience. Before every epoch, the data is shuffled.

## S3 HYPERPARAMETERS

The hyperparameters of different methods are given in Table S2 for *VizDoom* environment, in Table S3 for *DMLab* environment, and in Tables S4, S5 for *MuJoCo* Ant environment. The hyperparameters for *DMLab* are tuned on the "Sparse" environment for all methods — because all methods work reasonably on this environment (it is unfair to tune a method on an environment where it fails and also unfair to tune different methods on different environments). We use the PPO algorithm from the open-source implementation[9]. For implementation convenience, we scale both the bonus and the task reward (with a single balancing coefficient it would not be possible to turn off one of those rewards).

Table S2: Hyper-parameters used for *VizDoom* environment.

| | PPO | PPO + ICM | PPO + EC |
|---|---|---|---|
| Learning rate | 0.00025 | 0.00025 | 0.00025 |
| PPO entropy coefficient | 0.01 | 0.01 | 0.01 |
| Task reward scale | 5 | 5 | 5 |
| Curiosity bonus scale $\alpha$ | 0 | 0.01 | 1 |
| ICM forward inverse ratio | - | 0.2 | - |
| ICM curiosity loss strength | - | 10 | - |
| EC memory size | - | - | 200 |
| EC reward shift $\beta$ | - | - | 0.5 |
| EC novelty threshold $b_{novelty}$ | - | - | 0 |
| EC aggregation function $F$ | - | - | percentile-90 |

---

[9] https://github.com/openai/baselines

Table S3: Hyper-parameters used for *DMLab* environment.

| | PPO | PPO + ICM | PPO + Grid Oracle | PPO + EC |
|---|---|---|---|---|
| Learning rate | 0.00019 | 0.00025 | 0.00025 | 0.00025 |
| PPO entropy coefficient | 0.0011 | 0.0042 | 0.0066 | 0.0021 |
| Task reward scale | 1 | 1 | 1 | 1 |
| Curiosity bonus scale $\alpha$ | 0 | 0.55 | 0.052 | 0.030 |
| Grid Oracle cell size | - | - | 30 | - |
| ICM forward inverse ratio | - | 0.96 | - | - |
| ICM curiosity loss strength | - | 64 | - | - |
| EC memory size | - | - | - | 200 |
| EC reward shift $\beta$ | - | - | - | 0.5 |
| EC novelty threshold $b_{novelty}$ | - | - | - | 0 |
| EC aggregation function $F$ | - | - | - | percentile-90 |

Table S4: Hyper-parameters used for *MuJoCo* Ant "No Reward" environment. For the PPO+1 baseline, the curiosity reward is substituted by $+1$ (optimizes for survival). The curiosity bonus scale is applied to this reward.

| | PPO | PPO+1 | PPO + EC |
|---|---|---|---|
| Learning rate | 0.0003 | 0.00007 | 0.00007 |
| PPO entropy coefficient | 8e-6 | 0.0001 | 0.00002 |
| Task reward scale | 0 | 0 | 0 |
| Curiosity bonus scale $\alpha$ | 0 | 1 | 1 |
| EC memory size | - | - | 1000 |
| EC reward shift $\beta$ | - | - | 1 |
| EC novelty threshold $b_{novelty}$ | - | - | 0 |
| EC aggregation function $F$ | - | - | 10th largest |

Table S5: Hyper-parameters used for *MuJoCo* Ant "Escape Circle" environment. For the PPO+1 baseline, the curiosity reward is substituted by $+1$ (optimizes for survival). The curiosity bonus scale is applied to this reward.

| | PPO | PPO+1 | PPO + EC |
|---|---|---|---|
| Learning rate | 0.0001 | 0.0001 | 4.64e-05 |
| PPO entropy coefficient | 1.21e-06 | 1.43e-06 | 1.78e-06 |
| Task reward scale | 1 | 1 | 1 |
| Curiosity bonus scale $\alpha$ | 0 | 0.85 | 0.25 |
| EC memory size | - | - | 1000 |
| EC reward shift $\beta$ | - | - | 1 |
| EC novelty threshold $b_{novelty}$ | - | - | 0 |
| EC aggregation function $F$ | - | - | 10th largest |

## S4    R-NETWORK GENERALIZATION STUDY

One of the promises of our approach is its potential ability to generalize between tasks. In this section we verify if this promise holds.

### S4.1    TRAINING R-NETWORK ON ALL *DMLab*-30 TASKS

Could we train a universal R-network for all available levels — and then use this network for all our tasks of interest? Since different games have different dynamics models, the notion of closely

reachable or far observations also changes from game to game. Can R-network successfully handle this variability? Table S6 suggests that using a universal R-network slightly hurts the performance compared to using a specialized R-network trained specifically for the task. However, it still definitely helps to get higher reward compared to using the plain PPO. The R-network is trained using 10M environment interactions equally split across all 30 *DMLab*-30 tasks.

Table S6: Reward on the tasks "No Reward" and "Very Sparse" using a universal R-network. Two baselines (PPO and PPO + EC with a specialized R-network) are also provided.

| Method | No Reward | Very Sparse |
|---|---|---|
| PPO | $191 \pm 12$ | $8.6 \pm 4.3$ |
| PPO + EC with specialized R-network | $475 \pm 8$ | $24.7 \pm 2.2$ |
| PPO + EC with universal R-network | $348 \pm 8$ | $19.3 \pm 1.0$ |

## S4.2 TRAINING R-NETWORK ON ONE LEVEL AND TESTING ON ANOTHER

This experiment is similar to the previous one but in a sense is more extreme. Instead of training on all levels (including the levels of interest and other unrelated levels), can we train R-network on just one task and use if for a different task? Table S7 suggests we can obtain reasonable performance by transferring the R-network between similar enough environments. The performance is unsatisfactory only in one case (using the R-network trained on "Dense 2"). Our hypothesis is that the characteristics of the environments are sufficiently different in that case: single room versus maze, static textures on the walls versus changing textures.

Table S7: Reward on the environments "No Reward" and "Very Sparse" (columns) when the R-network is trained on different environments (rows). We provide a result with a matching R-network for reference (bottom).

| R-network training environment | No Reward | Very Sparse |
|---|---|---|
| Dense 1 | $320 \pm 5$ | $18.5 \pm 1.4$ |
| Dense 2 | $43 \pm 2$ | $0.8 \pm 0.5$ |
| Sparse + Doors | $376 \pm 7$ | $16.2 \pm 0.7$ |
| Matching environment | $475 \pm 8$ | $24.7 \pm 2.2$ |

## S5 STABILITY/ABLATION STUDY

The experiments are done both in "No Reward" and "Very Sparse" environments. The "No Reward" environment is useful to avoid the situations where task reward would hide important behavioural differences between different flavors of our method (this "hiding" effect can be easily observed for different methods comparison in the dense reward tasks — but the influence of task reward still remains even in sparser cases). As in the main text, for the "No Reward" task we report the Grid Oracle reward as a discrete approximation to the area covered by the agent trajectories.

### S5.1 POSITIVE EXAMPLE THRESHOLD IN R-NETWORK TRAINING

Training the R-network requires a threshold $k$ to separate negative from positive pairs. The trained policy implicitly depends on this threshold. Ideally, the policy performance should not be too sensitive to this hyper-parameter. We conduct a study where the threshold is varied from 2 to 10 actions (as in all experiments before, each action is repeated 4 times). Table S8 shows that the EC performance is reasonably robust to the choice of this threshold.

Table S8: Reward in the "No Reward" and "Very Sparse" tasks using different positive example thresholds $k$ when training the R-network.

| Threshold $k$ | No Reward | Very Sparse |
|---|---|---|
| 2 | $378 \pm 18$ | $28.3 \pm 1.6$ |
| 3 | $395 \pm 10$ | $20.9 \pm 1.6$ |
| 4 | $412 \pm 8$ | $31.1 \pm 1.2$ |
| 5 | $475 \pm 8$ | $24.7 \pm 2.2$ |
| 7 | $451 \pm 4$ | $23.6 \pm 1.0$ |
| 10 | $455 \pm 7$ | $20.8 \pm 0.8$ |

## S5.2 MEMORY SIZE IN EC MODULE

The EC-module relies on an explicit memory buffer to store the embeddings of past observations and define novelty. One legitimate question is to study the impact of the size of this memory buffer on the performance of the EC-module. As observed in table S9, the memory size has little impact on the performance.

Table S9: Reward for different values of the memory size for the tasks "No Reward" and "Very Sparse".

| Memory size | No Reward | Very Sparse |
|---|---|---|
| 100 | $447 \pm 6$ | $19.4 \pm 1.9$ |
| 200 | $475 \pm 8$ | $24.7 \pm 2.2$ |
| 350 | $459 \pm 6$ | $23.5 \pm 1.4$ |
| 500 | $452 \pm 6$ | $23.8 \pm 2.0$ |

## S5.3 ENVIRONMENT INTERACTION BUDGET FOR TRAINING R-NETWORK

The sample complexity of our EC method includes two parts: the sample complexity to train the R-network and the sample complexity of the policy training. In the worst case – when the R-network does not generalize across environments – the R-network has to be trained for each environment and the total sample complexity is then the sum of the previous two sample complexities. It is then crucial to see how many steps are needed to train R-network such that it can capture the notion of reachability. R-network trained using a number of environment steps as low as 1M already gives good performance, see Table S10.

Table S10: Reward of the policy trained on the "No Reward" and "Very Sparse" tasks with an R-network trained using a varying number of environment interactions (from 100K to 5M).

| Interactions | No Reward | Very Sparse |
|---|---|---|
| 100K | $357 \pm 18$ | $12.2 \pm 1.3$ |
| 300K | $335 \pm 9$ | $16.2 \pm 0.7$ |
| 1M | $383 \pm 13$ | $18.6 \pm 0.9$ |
| 2.5M | $475 \pm 8$ | $24.7 \pm 2.2$ |
| 5M | $416 \pm 5$ | $20.7 \pm 1.4$ |

## S5.4 IMPORTANCE OF TRAINING DIFFERENT PARTS OF R-NETWORK

The R-network is composed of an Embedding network and a Comparator network. How important is each for the final performance of our method? To establish that, we conduct two experiments. First, we fix the Embedding network at the random initialization and train only the Comparator. Second, we substitute the Comparator network applied to embeddings $\mathbf{e}_1, \mathbf{e}_2$ with the sigmoid function

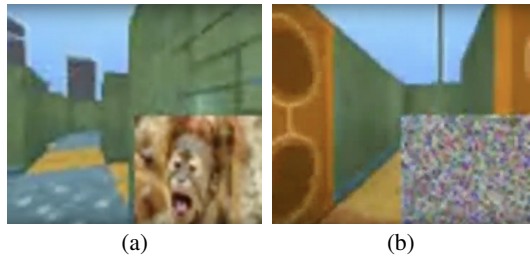

(a)                    (b)

Figure S1: Examples of randomized environments: (a) Image Action, (b) Noise.

$\sigma(\mathbf{e}_1^T \mathbf{e}_2)$ and train only the Embedding. According to the results in Table S11, we get a reasonable performance with a random embedding: the results are still better than the plain PPO (but worse than with the complete R-network). However, without the Comparator the quality drops below the plain PPO.

This experiment leads us to two conclusions. First, training the Embedding network is desired but not necessary for our method to work. Second, using the Comparator is essential and cannot be omitted in the current setup. Apparently, predicting reachability requires fine-grained access to both embeddings at the same time — and a simple comparison function does not work.

Table S11: Reward on the "No Reward" and "Very Sparse" tasks using ablated versions of the R-network.

| Method | No Reward | Very Sparse |
|---|---|---|
| PPO | $191 \pm 12$ | $8.6 \pm 4.3$ |
| PPO + EC with complete R-network | $475 \pm 8$ | $24.7 \pm 2.2$ |
| PPO + EC with random Embedding | $392 \pm 12$ | $16.2 \pm 1.4$ |
| PPO + EC without Comparator network | $48 \pm 3$ | $5.8 \pm 2.4$ |

## S6    RANDOMIZED ENVIRONMENTS

In the main text of the paper we observed how the firing action confused the surprise-based curiosity method ICM. This was a manifestation of the hardness of future prediction performed by ICM. Importantly, there could be more than one reason why future prediction is hard (as observed in the concurrent work (Burda et al., 2018b)): partial observability of the environment, insufficiently rich future prediction model or randomized transitions in the environment. Since our own method EC relies on comparisons to the past instead of predictions of the future, one could expect it to be more robust to those factors (intuitively, comparison to the past is an easier problem). The goal of this section is to provide additional evidence for that.

We are going to experiment with one source of future prediction errors which we have used in the thought experiment from the introduction: environment stochasticity. In particular, we analyze how different methods behave when all the states in the environment provide stochastic next state. For that, we create versions of the *DMLab* environments "Sparse" and "Very Sparse" with added strong source of stochasticity: randomized TV on the head-on display of the agent. It is implemented as follows: the lower right quadrant of the agent's first person view is occupied with random images. We try a few settings:

- "Image Action $k$": there are $k$ images of animals retrieved from the internet, an agent has a special action which changes an image on the TV screen to a random one from this set. An example is shown in Figure S1(a).
- "Noise": at every step a different noise pattern is shown on the TV screen, independently from agent's actions. The noise is sampled uniformly from $[0, 255]$ independently for each pixel. An example is shown in Figure S1(b).
- "Noise Action": same as "Noise", but the noise pattern only changes if the agent uses a special action.

Table S12: Reward in the randomized-TV versions of *DMLab* task "Sparse" (mean $\pm$ std) for all compared methods. Higher is better. "Original" stands for the non-randomized standard version of the task which we used in the main text. "ECO" stands for the online version of our method, which trains R-network and the policy at the same time. The Grid Oracle method is given for reference — it uses privileged information unavailable to other methods. Results are averaged over 30 random seeds. No seed tuning is performed.

| Method | Image Action | | | Noise | Noise Action | Original |
|---|---|---|---|---|---|---|
| | 3 | 10 | 30 | | | |
| PPO | $11.5 \pm 2.1$ | $10.9 \pm 1.8$ | $8.5 \pm 1.5$ | $11.6 \pm 1.9$ | $9.8 \pm 1.5$ | $27.0 \pm 5.1$ |
| PPO + ICM | $10.0 \pm 1.2$ | $10.5 \pm 1.2$ | $6.9 \pm 1.0$ | $7.7 \pm 1.1$ | $7.6 \pm 1.1$ | $23.8 \pm 2.8$ |
| PPO + EC (ours) | $19.8 \pm 0.7$ | $15.3 \pm 0.4$ | $13.1 \pm 0.3$ | $18.7 \pm 0.8$ | $14.8 \pm 0.4$ | $26.2 \pm 1.9$ |
| PPO + ECO (ours) | $\mathbf{24.3 \pm 2.1}$ | $\mathbf{26.6 \pm 2.8}$ | $\mathbf{18.5 \pm 0.6}$ | $\mathbf{28.2 \pm 2.4}$ | $\mathbf{18.9 \pm 1.9}$ | $\mathbf{41.6 \pm 1.7}$ |
| PPO + Grid Oracle | $37.7 \pm 0.7$ | $37.1 \pm 0.7$ | $37.4 \pm 0.7$ | $38.8 \pm 0.8$ | $39.3 \pm 0.8$ | $56.7 \pm 1.3$ |

Table S13: Reward in the randomized-TV versions of *DMLab* task "Very Sparse" (mean $\pm$ std) for all compared methods. Higher is better. "Original" stands for the non-randomized standard version of the task which we used in the main text. "ECO" stands for the online version of our method, which trains R-network and the policy at the same time. The Grid Oracle method is given for reference — it uses privileged information unavailable to other methods. Results are averaged over 30 random seeds. No seed tuning is performed.

| Method | Image Action | | | Noise | Noise Action | Original |
|---|---|---|---|---|---|---|
| | 3 | 10 | 30 | | | |
| PPO | $6.5 \pm 1.6$ | $8.3 \pm 1.8$ | $6.3 \pm 1.8$ | $8.7 \pm 1.9$ | $6.1 \pm 1.8$ | $8.6 \pm 4.3$ |
| PPO + ICM | $3.8 \pm 0.8$ | $4.7 \pm 0.9$ | $4.9 \pm 0.7$ | $6.0 \pm 1.3$ | $5.7 \pm 1.4$ | $11.2 \pm 3.9$ |
| PPO + EC (ours) | $13.8 \pm 0.5$ | $10.2 \pm 0.8$ | $7.4 \pm 0.5$ | $13.4 \pm 0.6$ | $11.3 \pm 0.4$ | $24.7 \pm 2.2$ |
| PPO + ECO (ours) | $\mathbf{20.5 \pm 1.3}$ | $\mathbf{17.8 \pm 0.8}$ | $\mathbf{16.8 \pm 1.4}$ | $\mathbf{26.0 \pm 1.6}$ | $\mathbf{12.5 \pm 1.3}$ | $\mathbf{40.5 \pm 1.1}$ |
| PPO + Grid Oracle | $35.4 \pm 0.6$ | $35.9 \pm 0.6$ | $36.3 \pm 0.7$ | $35.5 \pm 0.6$ | $35.4 \pm 0.8$ | $54.3 \pm 1.2$ |

The results at 20M 4-repeated environment steps are shown in Tables S12, S13. In almost all cases, the performance of all methods deteriorates because of any source of stochasticity. However, our method turns out to be reasonably robust to all sources of stochasticity and still outperforms the baselines in all settings. The videos[10,11] demonstrate that our method still explores the maze reasonably well.

## S7 COMPUTATIONAL CONSIDERATIONS

The most computationally intensive parts of our algorithm are the memory reachability queries. Reachabilities to past memories are computed in parallel via mini-batching. We have shown the algorithm to work reasonably fast with a memory size of 200. For orders of magnitude larger memory sizes, one would need to better parallelize reachability computations — which should in principle be possible. Memory consumption for the stored memories is very modest (400 KB), as we only store 200 of 512-float-embeddings, not the observations.

As for the speed comparison between different methods, PPO + ICM is 1.09x slower than PPO and PPO + EC (our method) is 1.84x slower than PPO. In terms of the number of parameters, R-network brings 13M trainable variables, while PPO alone was 1.7M and PPO + ICM was 2M. That said, there was almost no effort spent on optimizing the pipeline in terms of speed/parameters, so it is likely easy to make improvements in this respect. It is quite likely that a resource-consuming Resnet-18 is

---

[10]Image Action: https://youtu.be/UhF1MmusIU4
[11]Noise: https://youtu.be/4B8VkPA2Mdw

not needed for the R-network — a much simpler model may work as well. In this paper, we followed the setup for the R-network from prior work (Savinov et al., 2018) because it was shown to perform well, but there is no evidence that this setup is necessary.

## S8    ADDITIONAL *DMLab* TRAINING CURVES

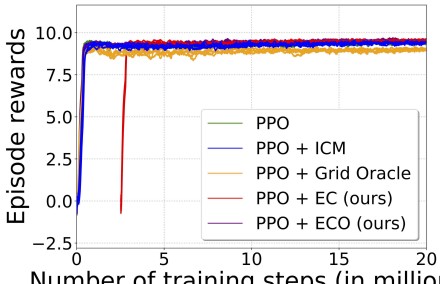

Figure S2:  Reward as a function of training step for the *DMLab* task "Dense 2".  Higher is better. We shift the curves for our method by the number of environment steps used to train R-network — so the comparison between different methods is fair.  We run every method 30 times and show 5 randomly selected runs.  No seed tuning is performed.

We show additional training curves from the main text experimental section in Figure S2.

