# OpenReview forum: "Episodic Curiosity through Reachability"
_ICLR.cc/2019/Conference_

### Official Review · AnonReviewer1 · 2018-11-02

**Rating:** 8
**Confidence:** 3

**Review:**

The authors propose an exploration bonus that is aimed to aid in sparse reward RL problems. The bonus is given by an auxillary network which tries to score whether a candidate observation is difficult to reach with respect to all previously observed novel observations which are stored in a memory buffer. The paper considers many experiments on complex 3D environments.

The paper is well written and very well illustrated. The method can be clearly understood from the 3 figures and the examples are nice. I think the method is interesting and novel and it is evaluated on a realistic and challenging problem.

It would be good if the authors could further elaborate on the scalability of the method in terms of compute/memory requirements and related to that if the implementation is cumbersome. I didn’t understand well how the method avoids the issue of old memories leaving the buffer. It seems for a large enough environment important observations will eventually become discarded causing a poor approximation of the curiosity bonus? For the large scale experiments I would like to know more rough details of the number of the compute time needed for the method relative to the PPO baseline and the other baseline (e.g. number of nodes for example and how long they run approximately)

Are there any potential issues with adapting the method on 2D environments like Atari? this could permit direct comparisons with several other recently proposed techniques in this area.

The Grid-Oracle result is very interesting and a contribution on it’s own if similar results for complex 3D environments are not published anywhere else. It demonstrates well that exploration bonuses can help drastically in these tasks. I think if possible it would be interesting to have an idea how fast this method converges (number of training steps) and not just the final reward as reported in the tables. Indeed as a general problem the current number of training steps of any methods shown seem to indicate these techniques are too data hungry for non-simulated environments. For some applications (e.g. aimed at sim-to-real transfer) the grid-oracle approach might be a good alternative to consider. I would be interested to know if the authors had some thoughts on this.

Overall I lean towards accept, the method is shown to work on relatively very complex problems in DMLab and VizDoom while most sparse reward solutions proposed are typically evaluated on relatively simpler and unrealistic tasks. I would consider to further increase my score if the authors can address some of the comments.

---

> ### Author Response · Authors · 2018-11-10
> **Authors' response**
>
> We thank the reviewer for their work on the review.
>
> > It would be good if the authors could further elaborate on the scalability of the method in terms of compute/memory requirements
>
> The most computationally intensive parts of the algorithm are the memory reachability queries. Reachabilities to past memories are computed in parallel via mini-batching. We have shown the algorithm to work reasonably fast with a memory size of 200. For significantly larger memory sizes, one would need to better parallelize reachability computations -- which should be doable. Memory consumption for the stored memories is very modest (400 KB), as we only store 200 of 512-float-embeddings, not the observations.
>
> > and related to that if the implementation is cumbersome.
>
> The implementation is relatively easy. We commit to publishing the source code if the paper is accepted -- this would make adoption even easier.
>
> > I didn’t understand well how the method avoids the issue of old memories leaving the buffer.
>
> Forgetting is unavoidable if the storage size is limited. That said, not all old memories are erased. The distribution of memory age is geometric: so older memories are sparser than the recent ones, but still present. Please see our visualization of the memory state: https://youtu.be/mphIRR6VsbM. Please note that we denote memories by their location only for visualization purposes, the coordinates are not available to our method.
>
> > It seems for a large enough environment important observations will eventually become discarded causing a poor approximation of the curiosity bonus?
>
> This is true: when revisiting a part of state space that the agent hasn’t been to for a long time, many of the memories from that region may have been discarded and the curiosity bonus may offer more reward for returning to these states. This should not be a problem though: the curiosity bonus would still provide some reactive incentive to move away from recent memories -- because recent memories are always well-represented. And, it is possible that it would be good to incentivise visiting states that haven’t been seen in a long time.
>
> > For the large scale experiments I would like to know more rough details of the number of the compute time needed for the method relative to the PPO baseline and the other baseline (e.g. number of nodes for example and how long they run approximately)
>
> PPO+ICM is 1.09x slower than PPO and PPO+EC (our method) is 1.84x slower than PPO. As for the number of parameters, R-network brings 13M trainable variables, while PPO alone was 1.7M and PPO+ICM was 2M. That said, we have spent almost no effort optimizing it in terms of speed/parameters, so it is likely easy to make improvements in this respect. It’s quite likely that we do not need a Resnet-18 for the R-network -- a much simpler model may work as well. In this paper, we just followed the setup for the R-network from prior work https://arxiv.org/abs/1803.00653 because it was shown to perform well, but there is no evidence that this setup is necessary.
>
> > Are there any potential issues with adapting the method on 2D environments like Atari? this could permit direct comparisons with several other recently proposed techniques in this area.
>
> We haven't tried it for Atari, so it is hard to predict. That said, we try to focus on more visually complex environments. In Atari, there is always a danger that the method would exploit exact observation repeatability. One recent work https://arxiv.org/pdf/1606.04460.pdf estimated this repeatability to reach 60% in some games, and > 10% in many. This creates a dangerous incentive for the exploration algorithms to brute-force this vulnerability. On the other hand, in DMLab, such repeatability was estimated by the same work as < 0.1%.
>
> > The Grid-Oracle result is very interesting and a contribution on it’s own… I think if possible it would be interesting to have an idea how fast this method converges (number of training steps)
>
> We don’t include Grid-Oracle into the plots because otherwise it is hard to see the difference between the comparable methods (note that it is not fair to compare Oracle with them). That said, Oracle converges faster than any other method -- but requires privileged information. To give specific numbers, after 5M 4-repeated steps Grid-Oracle reaches approximately reward 40 in the "Sparse" environment, reward 35 in the "Very Sparse" environment and reward 20 in the "Sparse+Doors" environment. This is way higher than any other method in our study. We will include those numbers into the manuscript.
>
> > For some applications (e.g. aimed at sim-to-real transfer) the grid-oracle approach might be a good alternative to consider.
>
> The Oracle could be useful in situations where additional information is available about the environment. However, it is not universal, so we have not focused on the possibility of taking advantage of privileged information in the current manuscript.

---

> > ### Comment · AnonReviewer1 · 2018-11-23
> > **reply**
> >
> > Thanks for your response. You have addressed my main concerns and have also added new results. I have increased my score as I think the paper is rather polished and well above the bar for acceptance at ICLR.
> >
> > I encourage the authors to integrate some of their discussions regarding scalability in the manuscript

---

> > > ### Author Response · Authors · 2018-11-23
> > > **Authors’ response**
> > >
> > > Thank you for your encouraging feedback! We will include the discussions into the camera-ready version of the paper.

---

### Official Review · AnonReviewer3 · 2018-11-02
**Not enough motivation why crustily driven approach is in interest.**

**Rating:** 6
**Confidence:** 4

**Review:**

In this paper, the authors study the problem of exploration in RL when the reward process is sparse. They introduce a new curiosity based approach which considers a state novel if it was not visited before and is far from the visited states. They show that their methods perform better than two other approaches, one without curiosity-driven exploration and the second one a one-step curiosity-driven approach.

The paper is well-written and easy to follow. The authors motivate this work by bringing an example where the state observation might be novel but important. They show that if part of the environment just changes randomly, then there is no need to explore there as much as vanilla curiosity-driven approaches to suggest. The approach in this paper partially addresses this drawback of curiosity-driven approaches. The authors miss the point why the curiosity-driven exploration approaches as in this work are in interest.

The problem mentioned in this paper can be also solved based on efficient exploration-exploration methods where the distribution of next states is considered rather than the samples themselves. An efficient explorative/exploitative RL agent explores part of state space more if there is uncertainty in the reward and state distribution rather than not being able to predict a particular sample. In curiosity-driven approaches, if the predictability of the next state is considered, all methods are sentenced to failure in stochastic environments. The approach in this paper partially mitigate this problem but for a very specific environment setup, but still, fails if the environment is stochastic.

---

> ### Author Response · Authors · 2018-11-06
> **Authors' response**
>
> > The authors miss the point why the curiosity-driven exploration approaches as in this work are in interest.
>
> It would be helpful if the reviewer could be more specific here.
>
> > The problem mentioned in this paper can be also solved based on efficient exploration-exploration methods where the distribution of next states is considered rather than the samples themselves.
>
> Could the reviewer please provide references to such methods, demonstrated on visually-rich 3D environments?
>
> > In curiosity-driven approaches, if the predictability of the next state is considered, all methods are sentenced to failure in stochastic environments. The approach in this paper partially mitigate this problem but for a very specific environment setup,
>
> ViZDoom and DMLab are standard benchmarks. We used the standard action sets for those benchmarks. Could the reviewer please elaborate more on what is very specific about our environmental setup?
>
> > but still, fails if the environment is stochastic.
>
> Could the reviewer please be more specific here? That is, how does the method fail in the case that the environment is stochastic?

---

> > ### Comment · AnonReviewer3 · 2018-11-07
> > **Clarification**
> >
> > I agree there are quite a few papers about curiosity-driven approaches, but they are mainly heuristic approaches for deterministic settings. I would like the authors to motivate and clarify why they use this approach in the stochastic setting. The problem set up (couch-potato) to motivate the approach in this paper is not general enough. What if all the states provide stochastic next state? Then the current method breaks? The curiosity methods extend to the stochastic settings if the curiosity is derived based on distribution mismatch, if it is not, then as the authors also mentioned it results in the couch-potato problem.
> >
> > I agree that the authors put effort for their empirical study and showed improvement. But I am not sure the algorithmic idea behind this work provides sufficient contribution. I am willing to change my score if the authors can address these.

---

> > > ### Author Response · Authors · 2018-11-10
> > > **Authors' response**
> > >
> > > We thank the reviewer for their work on the review and their clarification answer.
> > >
> > > First, we would like to point out that the stochastic environments are not a focus of our work. Couch-potato behaviour is not unique to stochastic environments. As we show in our experiments, perfectly normal deterministic environments could lead to such behaviour: partial observability or just hardness of future prediction can confuse the surprise-based ICM method (which was chosen as a baseline because it showed state-of-the-art results in visually-rich 3D environment ViZDoom in the prior work https://arxiv.org/pdf/1705.05363.pdf). The randomized TV example is explicitly labeled as a "thought example" in our paper. It was chosen for the sake of illustration. We will clarify this in the paper.
> > >
> > > Second, our method can work in an environment where "all the states provide stochastic next state". We created a version of our “VerySparse” environment with a randomized TV on the head-on display. More precisely, the lower right quadrant of the first-person view is occupied with an image from a set of 10 images. The change of the image on the TV is initiated by an additional action, provided to the agent. Using this action leads to a random image from the set to be shown on the TV. Our method still works in this setting: https://youtu.be/UhF1MmusIU4 (preliminary result computed at 7M 4-repeated steps). Additionally, we tried showing random noise on the TV screen at every step -- and our method works there as well: https://youtu.be/4B8VkPA2Mdw.

---

### Official Review · AnonReviewer2 · 2018-11-03
**A simple novel idea for improving exploration in DRL**

**Rating:** 8
**Confidence:** 4

**Review:**

The main idea of this paper is to propose a heuristic method for exploration in deep reinforcement learning. The work is fairly innovative in its approach, where an episodic memory is used to store agent’s observations while rewarding the agent for reaching novel observations not yet stored in memory. The novelty here is determined by a pre-trained network that computes the within k-step-reachability of current observation to the observations stored in memory. The method is quite simple but promising and can be easily integrated with any RL algorithm.

They test their method on a pair of 3D environments, VizDoom and DMLab. The experiments are well executed and analysed.

Positives:
-	They do a rigorous analysis of parameters, and explicitly count the pre-training interactions with the environment in their learning curves.
-	This method does not hurt when dense environmental rewards are present.
-	The memory buffer is smaller than the episode length, which avoids trivial solutions.
-	The idea of having a discriminator assess distance between states is interesting.

Questions and critics:
-	The tasks explored in this paper are all navigation based tasks, would this method also apply equally successfully to non-navigation domains such as manipulation?
-	My main concern is that the pre-training of the embedding and comparator networks directly depends on how good the random exploration policy is that collects the data. In navigation domains it makes sense that the random policy could cover the space fairly well, however, this will not be the case for more complex tasks involving more complex dynamics.
-	It was surprising to me that the choice of k does not seem to be that important. As it implicitly defines what “novelty” means for an environment, I would have expected that its value should be calibrated better. Could that be a function of the navigation tasks considered?
-	The DMLab results are not great or comparable to the state-of-the-art methods, which may hinder interpreting how good the policies really are. This was perhaps a conscious choice given they are only interested in early training results, but that seems like a confound.
-	The architecture does not include an RNN which makes certain things very surprising even though they shouldn't (e.g. firing, or moving around a corner, are specifically surprising for ICM) as they cannot be learnt, but perhaps if they had an RNN in the architecture these would be easy to explain? Would be interesting to see what are the authors thoughts on this (apart from their computational complexity argument they mention)?
-	Having the memory contain only information about the current episode with no information transfer between episodes seems a bit strange to me, I would like to hear the motivation behind this?
-	The fact that the memory is reset between episodes, and that the buffer is small, can mean that effectively the method implements some sort of complex pseudo count over meta-states per episode?
-	The embedding network is only trained during the pre-training phase and frozen during the RL task. This sounds a bit limiting to me: what if the agent starts exploring part of the space that was not covered during pre-training? Obviously this could lead to collapses when allowing to fine-tune it, but I feel this is rather restrictive. Again, I feel that the choice of navigation tasks did not magnify this problem, which would arise more in harder exploration tasks.
-	I think that alluding that their method is similar to babies’ behaviour in their cradle is stretched at best and not a constructive way to motivate their work…
-	In Figure 6 and 7, all individual curves from each seed run are shown, which is a bit distracting. Perhaps showing the mean and std would be a cleaner and easier-to-interpret way to report these results?

Overall, it is a simple and interesting idea and seems quite easy to implement. However, everything is highly dependent on how varying the environment is, how bad the exploration policy used for pre-training is, how good the embeddings are once frozen, and how k, action repeat and memory buffer size interact. Given that the experiments are all navigation based, it makes it hard for me to assess whether this method can work as well in other domains with harder exploration setups.

---

> ### Author Response · Authors · 2018-11-10
> **Authors' response [part 1/2]**
>
> We thank the reviewer for their work on the review.
>
> > The tasks explored in this paper are all navigation based tasks, would this method also apply equally successfully to non-navigation domains such as manipulation?
>
> This is a great question and we work on experiments in other domains. However, we would like to point out that the tasks we already have in the paper are both non-trivial and more visually complex than in many other works in the field of sparse-reward exploration.
>
> > My main concern is that the pre-training of the embedding and comparator networks directly depends on how good the random exploration policy is that collects the data. In navigation domains it makes sense that the random policy could cover the space fairly well, however, this will not be the case for more complex tasks involving more complex dynamics.
>
> We agree that for more complex domains randomly collected data may be insufficient and online training of the R-network may be crucial. To address the concerns of the reviewer, we have implemented a version of our algorithm which performs training of R-network online (together with the policy). Preliminary results indicate that such training is possible and does not collapse. It produces results at least as good as pre-training. Thus, we are able to demonstrate that our approach can function with online training, offering the possibility of functioning in domains where collection of random data may be insufficient.
>
> In addition, we would like to point out that the R-network (Embedding + Comparator) can generalize beyond what was seen in the pre-training stage. We have such an experiment in the supplementary section S3 "R-network generalization study". In particular, in Table S4, R-networks trained on levels "Dense 1" and "Sparse + Doors" generalize to the "Very Sparse" environment. The visual gap is quite significant: please compare https://youtu.be/C5g10cUl7Ew with https://youtu.be/9J4CzdOz60I, for example. All this is possible because R-network is solving a simple problem of comparing two observations given access to both observations at the same time.
>
> Moreover, in the real world, people typically hand-design the initial exploration policy even for the standard RL methods, let alone the model-based ones (and our method could be considered partially model-based). For example, please take a look at the recent work https://ai.googleblog.com/2018/06/scalable-deep-reinforcement-learning.html (which has just received the best paper award at the Conference on Robotic Learning). Another recent work from ICLR’18 https://openreview.net/forum?id=BkisuzWRW also uses hand-crafted policy for the robotic manipulation task to collect data for training the inverse model of the environment.
>
> > It was surprising to me that the choice of k does not seem to be that important. As it implicitly defines what “novelty” means for an environment, I would have expected that its value should be calibrated better. Could that be a function of the navigation tasks considered?
>
> Those values of k are still rather small. What we demonstrate in this experiment is that our method is not excessively sensitive to this parameter when it is chosen within a reasonable range.
>
> > The DMLab results are not great or comparable to the state-of-the-art methods, which may hinder interpreting how good the policies really are. This was perhaps a conscious choice given they are only interested in early training results, but that seems like a confound.
>
> As far as we know, SOTA results are achieved by Impala https://arxiv.org/abs/1802.01561 at 1B steps (250M 4-repeated steps). We haven’t yet run our experiments at this scale: we use 20M 4-repeated steps in our PPO setup with 12 actors on GPU, which already takes 2 days to complete. Furthermore, being more sample efficient is an appealing property of more effective exploration as interactions with an environment might be costly in some environments.

---

> > ### Author Response · Authors · 2018-11-10
> > **Authors' response [part 2/2]**
> >
> > > The architecture does not include an RNN which makes certain things very surprising even though they shouldn't (e.g. firing, or moving around a corner, are specifically surprising for ICM) as they cannot be learnt, but perhaps if they had an RNN in the architecture these would be easy to explain? Would be interesting to see what are the authors thoughts on this (apart from their computational complexity argument they mention)?
> >
> > We agree it would be interesting to include an RNN into the architecture. As the reviewer mentions, doing so may help with certain kinds of surprising events. This would be worth exploring, but for interpretability of results and connection with past literature we have focused on feedforward architectures. An RNN was not a part of the original ICM approach to computing the reward bonus: the next-state prediction was done based on a few recent frames. In that sense, we followed the reference implementation -- which, as we verify, reproduces the published results. In the follow-up https://pathak22.github.io/large-scale-curiosity/resources/largeScaleCuriosity2018.pdf, the authors didn’t use an RNN in the policy either (personal communication with the authors).
> >
> > > Having the memory contain only information about the current episode with no information transfer between episodes seems a bit strange to me, I would like to hear the motivation behind this?
> >
> > The typical goal of RL is to maximize the reward throughout the current episode. The information from other episodes might be coming from a completely different environment/maze (unless you make an assumption that it is the same environment in every episode). If you visited some places in one maze, how would it help you to determine novelty of the current observation in another maze?
> >
> > > The fact that the memory is reset between episodes, and that the buffer is small, can mean that effectively the method implements some sort of complex pseudo count over meta-states per episode?
> >
> > Yes, it might be possible to understand the approach in this way. To gain a better understanding of how it works in practice, we created a visualization of the rewards, memory states and the trajectory of the agent during the episode. Please take a look here: https://youtu.be/mphIRR6VsbM. The distribution of states in memory is geometric: older memories are sparser but some are still there. This is enough to learn a reasonable exploration strategy in our environments.
> >
> > > The embedding network is only trained during the pre-training phase and frozen during the RL task. This sounds a bit limiting to me: what if the agent starts exploring part of the space that was not covered during pre-training? Obviously this could lead to collapses when allowing to fine-tune it, but I feel this is rather restrictive. Again, I feel that the choice of navigation tasks did not magnify this problem, which would arise more in harder exploration tasks.
> >
> > Please see our comment about online training and generalization above. We did not observe collapses in our new online training experiments, nor in most of our generalization experiments.
> >
> > > I think that alluding that their method is similar to babies’ behaviour in their cradle is stretched at best and not a constructive way to motivate their work…
> >
> > We will remove this inspiration.
> >
> > > In Figure 6 and 7, all individual curves from each seed run are shown, which is a bit distracting. Perhaps showing the mean and std would be a cleaner and easier-to-interpret way to report these results?
> >
> > We could re-do the plots if the reviewer wishes. However, we noticed some issues with the mean+-std kind of visualization as the distribution at each step is far from looking like a gaussian. In fact, it is clearly multimodal. For example, in Figure 6 it wouldn't be clear if mean < 1.0 means that the trained model doesn't always reach the goal or the training is unstable and some models reach the goal consistently while others fail consistently (the latter is actually the case for the baselines at some points during training).

---

> > > ### Author Response · Authors · 2018-11-12
> > > **Update on online training**
> > >
> > > After tuning online training of R-network, we obtained significantly improved results with respect to offline training: reward 26 -> 42 on Sparse in DMLab, reward 25 -> 41 on VerySparse in DMLab, reward 9 -> 20 on Sparse+Doors in DMLab. Also, results look qualitatively better now: offline training bumps into the walls quite often https://youtu.be/C5g10cUl7Ew -> online training almost doesn’t bump into the walls at all https://youtu.be/d2KiaWIJgfU.
> > >
> > > Thus, the experimental results justify that collecting data from a policy is ultimately a better way to train the R-network (probably because randomly visited states may be very unbalanced relative to what an agent actually encounters).
> > >
> > > Although the online training experiment was on our roadmap, we would like to thank the reviewer for motivating us to do it sooner rather than later! We will include the online training experiments in the paper.

---

> > > > ### Author Response · Authors · 2018-11-21
> > > > **Update on experiments in other domains**
> > > >
> > > > We have been able to learn Mujoco ant locomotion out of curiosity based on the first-person view: https://youtu.be/j_DToFnz9hQ (third-person view, for visualization only), https://youtu.be/8u_hbfEAo0w (first-person view, used by the curiosity module).
> > > >
> > > > First, let us describe the setup.
> > > > 1. Environment: the standard mujoco environment is a plane with uniform texture on it -- nothing to be visually curious about. To fix that, we tiled the 400x400 floor into squares of size 4x4. Each tile is assigned a random texture from a set of 190 textures at the beginning of every episode. The ant is initialized at the random location of the plane. The episode lasts for 1000 steps (no action repeat is used). If the center of mass of the ant is above or below the predefined threshold -- the episode ends prematurely (standard condition which is often used).
> > > > 2. Reward: curiosity reward for training and hyperparameter search, Grid Oracle reward for reporting results. Note: for the baselines we use Grid Oracle for hyperparameter search -- which puts them in a privileged position with respect to our method.
> > > > 3. Observation space: for computing the curiosity reward, we only use a first-person view camera mounted on the ant (that way we can use the same architecture of our curiosity module as in DMLab). For policy, we use the standard body features from Ant-v2 in gym/mujoco (joint angles, velocities, etc.).
> > > > 4. Action space: standard from Ant-v2 in gym/mujoco (continuous).
> > > > 5. Basic RL solver: PPO (same as before in the paper)
> > > > 6. Baselines: PPO on 0 reward (essentially random), PPO on constant +1 reward every step (optimizes for longer survival).
> > > >
> > > > Second, we present preliminary results. After 6M steps a random policy gives a Grid Oracle reward of 1.42, survival-PPO gives 2.19, our method achieves 3.95. Qualitatively, random policy dies quickly ( https://youtu.be/WFtM8-h8jOA ), survival-PPO survives for longer but does not move much ( https://youtu.be/b9ClgXOHpqA ), our method moves around the environment (first-person view: https://youtu.be/8u_hbfEAo0w , third-person view: https://youtu.be/j_DToFnz9hQ ). Average performance for our method is good, but not all random seeds produce good performance (results above are averaged over 3 seeds). We’re currently running more seeds with the best hyperparameters discovered for each method and investigating how training stability could be improved.
> > > >
> > > > Finally, let us discuss the relation to some other works in the field of learning locomotion from intrinsic reward that we are aware of (non-exhaustive and preliminary list). The closest work in terms of task setup is this concurrent ICLR submission https://openreview.net/forum?id=rJNwDjAqYX . The authors demonstrate slow motion of the ant ( https://youtu.be/l1FqtAHfJLI?t=90 ) learnt from pixel-based curiosity only. Other works use state features (like joint angles etc.) for formulating intrinsic reward, not pixels -- which is a different setup. One work in this direction is another concurrent ICLR submission https://openreview.net/forum?id=SJx63jRqFm .
> > > >
> > > > We hope this experiment addresses the valid concerns of the reviewer and demonstrates the generality of our method. We will include this experiment and the discussion into the camera-ready version of the paper.

---

> > > > > ### Comment · AnonReviewer2 · 2018-11-27
> > > > > **Thanks for your hard work**
> > > > >
> > > > > Thank you for your thorough response to my review!
> > > > >
> > > > > The experiments on online training of the R-network are very encouraging and I'm very glad that this resulted in improvements in performance.
> > > > >
> > > > > The extra MuJoCo ant locomotion experiments are interesting and I'm very much looking forward to reading the updated paper and seeing the final results of training in this task.
> > > > >
> > > > > I just want to point out that I'm very impressed by all the efforts made by the authors to address the comments raised in my review. They went above and beyond expected work in this rebuttal period!
> > > > >
> > > > > I believe the final version of the paper will be significantly stronger than the submitted version.
> > > > > Hence, I'm happy to increase my score to 8 after seeing the revised version of the manuscript.

---

> > > > > > ### Author Response · Authors · 2018-11-28
> > > > > > **Paper updated**
> > > > > >
> > > > > > Thank you for your help in improving our research! We have performed the paper update (the log is written in a separate message addressed to all the reviewers). We also added one more experiment with the MuJoCo Ant. Here is an anonymous link to the updated paper: https://drive.google.com/open?id=1tUHfBwWWu6W2zuk-De0AyYWxuNQTWa0D .

---

### Official Review · AnonReviewer4 · 2018-11-26
**Great idea, promising results, some confusing text**

**Rating:** 7
**Confidence:** 4

**Review:**

This paper proposes a new method to give exploration bonuses in RL algorithms by giving larger bonuses to observations that are farther away (> k) in environment steps to past observations in the current episode, encouraging the agent to visit observations farther away. This is in contrast to existing exploration bonuses based on prediction gain or prediction error, which do not work properly for stochastic transitions.

Overall, I very much like the idea, but I found many little pieces of confusing explanations that could be further clarified, and also some questionable implementation details. However the experimental results are very promising, and the approach should be modular and slotable into existing deep RL methods.

Section Introduction: I’m confused by how you can define such a bonus if the memory is the current episode. Won’t the shortest-path distance of the next observation always be 1 because it is immediately following the current step, and thus this results in a constant bonus? You explain how you get around this in practice, but intuitively and from a high-level, this idea does not make sense. It would perhaps make more sense if you used a different aggregation, such average, in which case you would be giving bonuses to observations that are farther away from the past on average.

Also, while eventually this idea makes sense, it only makes sense within a single episode. If you clear the memory between episodes, then you are relying on some natural stochasticity of the algorithm to avoid revisiting the same states as in the previous episode. Otherwise, it seems like there is not much to be gained from actually resetting and starting a new episode; it would encourage more exploration to just continue the same episode, or not clear memory when starting a new episode.

Section 2.2: You say you have a novelty threshold of 0 in practice, doesn’t this mean you end up always adding new observations to the memory? In this case, then it seems like your aggregation method of taking the 90th percentile is really the only mechanism that avoids the issue of always predicting a constant distance of 1 (and relying on the function approximator’s natural errors).

I do think you should rework your intuition. It seems to me what you are actually doing is creating some sort of implicit discretization of the observation space, and rewarding observations that you have not seen before under this discretization. This is what would correspond to a shortest-path distance aggregation.

Experiments: I like your grid oracle, as it acts as a baseline for using PPO and provides a point of reference for how well an exploration bonus could potentially be. But why aren’t grid oracle results put into your graphs? Your results look good and are very promising.

Other points:
- The pre-training of the R-network is concerning, but you have already responded with preliminary results.
- I do share some of the concerns other reviewers have brought up about generality beyond navigation tasks, e.g. Atari games. To me, it seems like this method can run into difficulty when reachability is not as nice as it is in navigation tasks, for example if the decisions of the task followed a more tree-like structure. This also does not work well with the fact that you reset every episode, so there is nothing to encourage an agent to try different branches of the tree every episode.

---

> ### Author Response · Authors · 2018-11-28
> **Authors’ response [part 1/2]**
>
> We thank the reviewer for their work on the review. Please note that the paper has just been updated as per the request from AR2, here is an anonymous link to the updated paper: https://drive.google.com/open?id=1tUHfBwWWu6W2zuk-De0AyYWxuNQTWa0D . The reviewer’s questions are addressed below.
>
> > Won’t the shortest-path distance of the next observation always be 1 because it is immediately following the current step, and thus this results in a constant bonus?
>
> In Section 2.2 we introduce a novelty threshold b_novelty for entering the memory buffer which addresses exactly this problem. This threshold implicitly discretizes the embedding space: only  sufficiently novel observations are rewarded with a bonus. Very likely, if the current observation is in memory, the next one won't be considered novel. Not only the next observation won't receive a large bonus, it also won't enter the memory buffer. As time passes by, the agent will go further and further away from observations in memory, the reward bonus will increase and at some point exceed b_novelty -- and only then the observation will enter the buffer. Of course, right after that the reward will drop again. Please take a look at this reward visualization: https://youtu.be/mphIRR6VsbM
>
> > It would perhaps make more sense if you used a different aggregation, such average, in which case you would be giving bonuses to observations that are farther away from the past on average.
>
> We have tried average in the past, it did not work well. The reason is probably that the average is not robust to outliers -- which are abundant as the visual similarity can't be perfect.
>
> > Also, while eventually this idea makes sense, it only makes sense within a single episode. If you clear the memory between episodes, then you are relying on some natural stochasticity of the algorithm to avoid revisiting the same states as in the previous episode. Otherwise, it seems like there is not much to be gained from actually resetting and starting a new episode; it would encourage more exploration to just continue the same episode, or not clear memory when starting a new episode.
>
> The typical goal of RL is to maximize the reward throughout the current episode. The information from other episodes might be coming from a completely different environment/maze (unless you make an assumption that it is the same environment in every episode). If you visited some places in one maze, how would it help you to determine novelty of the current observation in another maze?
>
> > Section 2.2: You say you have a novelty threshold of 0 in practice, doesn’t this mean you end up always adding new observations to the memory?
>
> As we have an additive factor beta = 0.5, the bonus b ends up in interval [-alpha/2, alpha/2]. Thus b_novelty = 0 is the middle of this interval and not all observations end up in the memory.
>
> > I do think you should rework your intuition. It seems to me what you are actually doing is creating some sort of implicit discretization of the observation space, and rewarding observations that you have not seen before under this discretization.
>
> This is exactly what we are doing -- by introducing b_novelty we implicitly discretize the embedding space.
>
> > I like your grid oracle, as it acts as a baseline for using PPO and provides a point of reference for how well an exploration bonus could potentially be. But why aren’t grid oracle results put into your graphs? Your results look good and are very promising.
>
> At the time of the submission, we didn’t include Grid-Oracle into the plots because otherwise it was hard to see the difference between the comparable methods (note that it is not fair to compare Oracle with them). In the latest version of our paper, we included it for all DMLab curves.

---

> > ### Author Response · Authors · 2018-11-28
> > **Authors' response [part 2/2]**
> >
> > > The pre-training of the R-network is concerning, but you have already responded with preliminary results.
> >
> > Now we have the final results for the online training of R-network, please take a look at the updated Table 1, "PPO + ECO" line. The improvement is quite significant. Moreover, we gained an improvement not only in sparse-reward tasks, but also in dense-reward ones. We hypothesise that our online curiosity model creates less contradiction between the actual task and being curious. This is maybe because the training data for R-network is sampled from the current policy which is solving the task. On the other hand, it is different for Oracle: there is no adaptation for the task at hand, so there is potentially more contradiction. In both Dense tasks "PPO + ECO" outperforms Grid Oracle (and plain PPO).
> >
> > > I do share some of the concerns other reviewers have brought up about generality beyond navigation tasks
> >
> > We have added a new experiment in the domain of learning locomotion out of curiosity. Please take a look at the supplementary section S1. The results look encouraging: a Mujoco ant learns to walk purely from maximizing our curiosity reward which is computed from the first-person view of the ant. Here are some videos: https://youtu.be/OYF9UcnEbQA (third-person view, for visualization only), https://youtu.be/klpDUdkv03k (first-person view, used by the curiosity module). We also have a comparison to some simple baselines in that section. Additionally, we have run an experiment with very sparse task reward: the ant is rewarded for escaping a large circle. In that task, our method also significantly outperforms the baselines.

---

### Public Comment · (anonymous) · 2018-10-25
**Mechanism Behind Generalization of Exploration?**

Excellent work! I was wondering if the authors would be able to provide some intuition for a phenomenon that I find a bit puzzling in this paper. I noticed that this algorithm is tested on a set of hold-out levels that are not seen during training. Given my understanding of the approach, I am not exactly sure why this would work at all. My understanding is that the proposed algorithm encourages exploration by rewarding the visitation of states that are distant from the ones that have been seen during each episode in training. It is sensible that this would encourage policies to learn to explore in the training environments, since they are receiving rewards for seeing rare states and learning to move towards those, but it is not clear why this would work in unseen environments. Why would policies that move towards "rare" states in the training set, work at all on a separate testing set? I suspect that perhaps the policies are learning some sort of generalized exploration behavior due to the wide variety of environments seen during training? It would be great if the authors could shed more light on this.

---

> ### Author Response · Authors · 2018-10-25
> **Authors' response**
>
> Thank you for your interest! Indeed, the policy learns generalized exploration behaviour, because it has seen approximately 11000 unique environments (differing in layouts and sets of textures) during 20M steps of training. All those environments are generated procedurally by the DMLab engine. Moreover, DMLab has a mechanism to ensure uniqueness and disjoint train/validation/test splits. However, please keep in mind that all those environments are still generated from some distribution implemented inside the DMLab engine.

---

### Author Response · Authors · 2018-11-10
**Response to the reviewers**

We thank the reviewers for their work and their valuable comments. We are happy that the reviewers find our method interesting and innovative (AR1, AR2), note that the paper is well-written and easy to understand (AR1, AR3), and mention that the experiments in our work are well-executed (AR1, AR2).

As for the reviewers’ questions, we would like to highlight two key points (including new interesting results):
1. Pretraining of R-network and its generalization: we further extended our work with online training, which is stable and gives significantly better results with respect to the pre-trained version. Moreover, we have evidence that the R-network generalizes well beyond the areas explored during training. It even generalizes between environments. This is because it "simply" needs to learn to meaningfully compare observations, not "recognize" observations. Please see our reply to AR2 for details.
2. Environment stochasticity: while it is not the focus of our work, we experimented with adding a strong source of stochasticity to the environment, and our method is reasonably robust to it. Please see our reply to AR3 for details.

We respond in detail to each of the reviewers individually in comments to their reviews. We will work on performing the proposed experiments and updating the paper accordingly.

---

### Author Response · Authors · 2018-11-23
**Authors’ rebuttal**

To summarize, we did the following experiments for the rebuttal:
1. (AR2) Online training of R-network. The results improved significantly with respect to offline training.
2. (AR2) Experiments in other domains: learning locomotion out of first-person-view curiosity for ant in Mujoco. Preliminary results demonstrate the applicability of our approach to this task.
3. (AR3) Experiments in stochastic environments. For all the settings we tried, our method still works.
4. (AR1, AR2) Reward/memory state visualization. Those videos provide more insight for the reader about how our method works.

We will include the experiments and discussions into the camera-ready version of the paper.

If there are any remaining questions/concerns, we would be grateful if the reviewers raise them so that we could further improve our work.

---

### Author Response · Authors · 2018-11-28
**Paper updated**

As requested by AR2, an update has been performed on the paper. As the system was already closed by the time we have received the request, here is an anonymous link to the updated paper: https://drive.google.com/open?id=1tUHfBwWWu6W2zuk-De0AyYWxuNQTWa0D .

The approximate log of the update:
1. Introduced online training into the main text of the paper, reported all final 20M-step DMLab results for it.
2. Announced Mujoco Ant experiments in the abstract, introduction, experimental setup section in the main paper. Added those experiments as the first section of the supplementary material (as there is no more space for them in the main text). We also conducted an additional experiment with very sparse reward -- our method shows good results there as well.
3. Added reward and memory state visualization video to the main text.
4. Removed the inspiration for R-network training.
5. Updated all experiments in the supplementary till the final 20M steps of training. We also added an ablation study which substitutes the Comparator network with a simpler function and establishes that the Comparator network is an essential part of our approach.
6. Added computational considerations section to the supplementary.
7. Updated DMLab curves with the online version of our algorithm and added Grid Oracle curves as well.
8. Added randomized TV experiments to the supplementary.

Dear reviewers, please note that this is a partial update because of the time constraints -- but we will perform all the other updates (which are more minor than the ones above) that we have promised in the camera-ready version.

---

### Meta-Review · Area_Chair1 · 2018-12-14
**Interesting idea with relevance to some common settings**

**Confidence:** 4
**Recommendation:** Accept (Poster)

**Metareview:**


The authors present a novel method for tackling exploration and exploitation that yields promising results on some hard navigation-like domains. The reviewers were impressed by the contribution and had some suggestions for improvement that should be addressed in the camera ready version.